biocomplexity/pattern recognition

multi-scale analysis, community detection, mobility patterns, COVID-19 exposure risk

**Author for correspondence:**
Leila Hedayatifar
e-mail: leila@necsi.edu

# Strategizing COVID-19 lockdowns using mobility patterns

Olha Buchel[1], Anton Ninkov[2], Danise Cathel[1], Yaneer Bar-Yam[1] and Leila Hedayatifar[1]

[1]New England Complex Systems Institute, 277 Broadway Street, Cambridge, MA, USA
[2]Faculty of Information and Media Studies, University of Western Ontario, Ontario, Canada

LH, 0000-0002-2185-5353

During the COVID-19 pandemic, governments have attempted to control infections within their territories by implementing border controls and lockdowns. While large-scale quarantine has been the most successful short-term policy, the enormous costs exerted by lockdowns over long periods are unsustainable. As such, developing more flexible policies that limit transmission without requiring large-scale quarantine is an urgent priority. Here, the dynamics of dismantled community mobility structures within US society during the COVID-19 outbreak are analysed by applying the Louvain method with modularity optimization to weekly datasets of mobile device locations. Our networks are built based on individuals' movements from February to May 2020. In a multi-scale community detection process using the locations of confirmed cases, natural break points from mobility patterns as well as high risk areas for contagion are identified at three scales. Deviations from administrative boundaries were observed in detected communities, indicating that policies informed by assumptions of disease containment within administrative boundaries do not account for high risk patterns of movement across and through these boundaries. We have designed a multi-level quarantine process that takes these deviations into account based on the heterogeneity in mobility patterns. For communities with high numbers of confirmed cases, contact tracing and associated quarantine policies informed by underlying dismantled community mobility structures is of increasing importance.

## 1. Introduction

The emergence and spread of the 2019 novel coronavirus (SARS-CoV-2 or COVID-19) has caused a global health emergency. With a high level of observed contagiousness [1] and an absence of proven medical treatments, the situation is increasingly dire.

Public health stakeholders race to find adequate methods for intervention as the outbreak spreads [2,3]. It is challenging to determine where the next outbreak will be and how to prevent or control it. Analysing data about positive tests and locations of current patients plays a critical role in public health agencies 'response' [4]. After the first cases came to the US through international travels, COVID-19 spread occurred rapidly through the population in patients both with or without symptoms at the time of transmission. COVID-19 has an incubation period that typically extends to 14 days, with a median time of 4–5 days [5].[1] Movement of asymptomatic individuals increases the risk of disease transmission in the public areas visited by them. So, it is important to define the geographical patches based on actual mobility of individuals in order to implement preventative policies with greater precision. In most cases, quarantine policies and data related to the COVID-19 outbreak are based on arbitrary borders such as state or county boundary lines [6]. In the US, state governments are responsible for the management and application of preventive policies inside their territories. For example, state governments closed public areas (e.g. work places, universities, schools and shopping centres), and asked people to wear masks. While state boundaries may serve constituents well in meeting certain social needs of their communities (e.g. infrastructure, taxes), they are not the most effective way to analyse data for anticipating disease outbreaks.

For the purposes of examining the spread of COVID-19 in the US, mobility patterns can be characterized in three overarching concepts: short distance (e.g. grocery shopping, walking), medium distance (e.g. travel for job or fun) and long distance (e.g. travel to other cities for vacation, visiting families). While short-distance movements lead to the local spread of the virus, medium and large-distance movements distribute the virus across larger scales to other cities and states. Travel can be thought of as occurring in 'bubbles' of progressively larger geographical scales. National travel bubbles include the collective movement of individuals travelling long distances from one region of the country to another. This type of mobility pattern was quickly identified as risky and attempts to limit it within the country were put in place. For example, at the very beginning of the North American outbreak in March 2020, a group of university students from around the country gathered on Florida beaches for uninhibited socialization during spring vacation, also known as 'Spring Break'. During this gathering, local transmission of COVID-19 was detected and the disease subsequently spread to other regions of the country long distances away from the original outbreak [7]. Soon after, most universities closed and airline travel was reduced. However, problematic bubbles of travel also form in local areas with close proximity due to more routine activities where there are more frequent and consistent mobility patterns. Local bubbles are prevalent in places such as the Northeast Megalopolis [8,9], where there are numerous cities and communities that all continuously connect to one another. In this region, many individuals live in one city/state (e.g. Philadelphia), work in another (New York City) and vacation in another (New Jersey coast). While these regions are separated by multiple administrative boundaries, they could still be considered to be in the same bubble.

The recent availability of large-scale human activity datasets has greatly improved our ability to study social systems [10–12]. Geo-located data sources enable direct observation of social interactions and collective behaviours with unprecedented detail. It is a well-established fact that aside from population heterogeneity, heterogeneity in movements at various distance scales has a large impact on the diffusion of infectious diseases [13]. Reductions in long-distance travel in the US due to lockdowns also showed a heterogeneous pattern at the county level [14]. Some mobility categories like transit and routine activities decreased in April 2020, which indicates a large negative per cent change during the first weeks of national lockdowns, and shows that people spent more time in residential areas [15]. On the other hand, some studies have shown that social isolation and hygiene had more impact on the reduction of positive active cases than lockdowns [16]. Networks of human mobility [17–19] have revealed the existence of geo-located communities or patches that exist at multiple scales from town to city, state and national scales [20,21]. People in these patches have similar movement patterns and, in a self-organized manner, mostly do not cross the borders of their communities. While the borders of mobility patterns of some patches follow administrative borders, the mobility patterns of other patches show a strong deviation from administrative geography [22].

Borders of patches are subject to vary during global events such as the COVID-19 pandemic. In this work, we study the dynamics of the modular structure of individual movements within the US in order to quantify effectiveness of policies that seek to lower transmission by reducing the amount of social movement and distance of travel within the population. We use anonymized cellphone data collected by SafeGraph company from February to May 2020 to build mobility networks based on the

---

[1]COVID-19 cases in states of the us. https://www.worldometers.info/coronavirus/country/us/.

movement of mobile device users between census block groups (CBGs). SafeGraph data have been widely used in detecting mobility patterns during the current pandemic. By applying the Louvain method as a community detection algorithm, fragmentation patterns of the networks are extracted. Patches represent the areas in which residents spend the most time. Applying the algorithm at multiple scales reveals the multi-scale modular structure of society and the features of that structure during the pandemic. We carefully study the properties and statistics of these patches and their dynamics in the presence of lockdown restrictions using the location of COVID-19 cases to define the risk of disease diffusion in mobility patches. Fusion of mobility and disease vectors has the potential to optimize future public health policies by restricting movements to and from infected patches, potentially reducing or eliminating the need for large-scale lockdowns.

# 2. Material and methods

## 2.1. Data

Mobility datasets: In March 2020, technology companies that gather geo-located information on individuals started to share anonymized mobility data to help researchers stop the spread of COVID-19. Here, we used the aggregated mobility datasets of the US by SafeGraph company to construct mobility networks from where individuals go. SafeGraph provides cellphone data and, for security of the users, anonymizes the data and aggregates them in CBGs. Each file describes individual CBGs and lists links with weights (number of links) to other CBGs that occurred on a specific day. First, we separate all these relationships and describe them as individual objects. Each relationship has a source, target, date and weight of interaction. Daily dataframes are combined into weekly dataframes; they are grouped and their relationships are summed. Each CBG in each relationship is augmented with central points derived from CBG polygons. See figure 1 for a summary of data statistics. We performed the analysis for weeks 23–29 February, 1–7 March, 8–14 March, 15–21 March, 22–28 March, 29 March–4 April, 5–11 April, 19–25 April, 26 April–2 May and 24–30 May.

COVID-19 datasets: We use daily confirmed cases time-series data from Johns Hopkins University COVID-19 Data Repository. This dataset provides cumulative counts of confirmed cases at county level for the US. By adding the number of active confirmed COVID-19 cases to the map, we define risk exposure for the communities.

## 2.2. Methods

To extract mobility fragmentation patterns, we first build mobility networks representing the connectivity of areas based on the movement of individuals. Applying a community detection algorithm shows us which geographical areas in the mobility networks are highly connected at multiple scales from mega-communities, to communities and sub-communities. To quantify changes in the network structures, we measure distributions of degree (number of movements from/to each CBG) and movement distances in communities.

Mobility network: In the mobility network, nodes represent a lattice with cells overlaid on a map of the US. Cells are CBGs used by SafeGraph and are the nodes in the mobility network. Edges represent the movement of an individual from one CBG (node) to another one. Edge's weight represents the number of people who travel between the two CBGs. This network aggregates the heterogeneities of human mobilities in a large-scale representation of social collective behaviours [23].

Community detection algorithm: We analyse social fragmentation by applying a community detection method to the mobility network. The term network fragmentation is often used to describe network dismantling in the literature [24,25]. In community detection algorithms such as the Girvan–Newman method [26], the term 'social fragmentation' is used to represent the modular structure of a social network in the absence of links. Communities refer to the regions in which nodes are connected to each other more than the rest of the network. Various community detection algorithms have been introduced for different purposes, accuracy and computing time [27,28]. In general, community detection methods can be categorized into two types: either agglomerative or divisive methods, which achieve optimization with modularity-based approaches like the Louvain [29] or fast-greedy algorithms [30], node similarity-based approaches like WalkTrap that uses a measure of distance based on a random walker [31], compression-based approaches that maximize compactness while minimizing information loss like InfoMod [32] and InfoMap [33] and others (see [27,28] for additional details and methods).

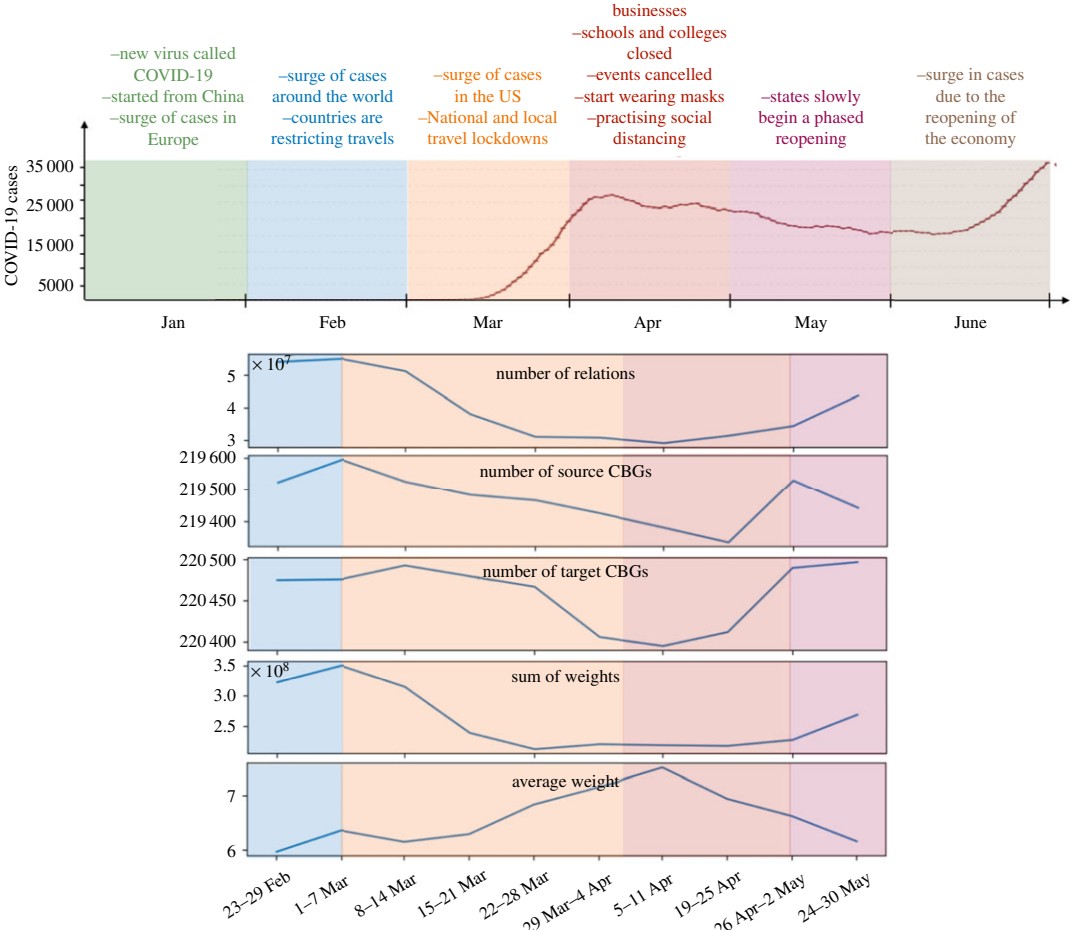

**Figure 1.** Top panel represents the US COVID-19 timeline and infected cases in the first half of 2020. Bottom panels represent summary of mobility network statistics during 10 weeks in February, March, April and May in 2020.

In this work, we use the Louvain algorithm [34] which works well for large networks and is relatively fast, and represents connected areas that deviate the most from a null model. The Louvain approach is an agglomerative method that considers each node as a single community in the first step. In an iterative process, nodes move to neighbouring communities and join them to maximize modularity [35]. In the next step, a network is built whose nodes are the communities in the previous step. The process is repeated in order to find the optimized value for modularity. Modularity is defined as $M = (1/2m)\sum_{ij}[A_{ij} - (k_i k_j/2m)]\delta(g_i, g_j)$. Where, $m$ represents the total number of links in the network, $A_{ij}$ counts link weight between nodes $i$ and $j$, and $k_i$ and $k_j$ are the sum of links to and from nodes $i$ and $j$. The second term in the equation indicates the expected number of links between the two nodes. $\delta(g_i, g_j)$ is equal to 1 if there is a link between the communities of node $i$ and $j$, otherwise it is 0. Modularity is a scalar value $-1 < M < 1$ that quantifies how distant the number of edges inside a community is from those of a random distribution. Values closer to 1 represent better detected communities. Due to the existence of multiple local minima in the Louvain algorithm, some variation in the assignment of nodes may occur between algorithm runs [35,36]. To quantify the stability of detected communities and identify areas in which communities overlap with each other, we generate an ensemble of multiple realizations and analyse the borders of patches in all the realizations [20,21].

Each CBG corresponds to a polygon and belongs to a community. We merge and dissolve polygons of CBGs that belong to the same community to represent communities as polygons. Rendering polygons with the same colour allows us to better visualize the communities and their borders on a map.

We define inter-community distances by creating a network that consists of communities as nodes and aggregate mobilities between them as links. Using this network structure we can further define clusters of communities with stronger connections. These clusters describe higher level aggregate behaviour enabling policy decisions about travel restrictions at the larger scale. We preserve colour

hues for mega-communities: purple hues were consistently assigned to northeastern states, blues to southeastern, greys to midwest, pink–magentas to southern communities and yellow–brown to the western communities. By applying the Louvain algorithm to the network inside communities, we define sub-communities in high population areas that mostly represent cities and surrounding metropolitan regions. Thus, our analysis shows three different scales of communities in the US.

Degree and distance distributions: We examined the distribution of the number of nodes versus degree (number of in and out links to the nodes) and number of links versus the length of the links using a survival function. If the degree of nodes/length of links per node/link is in range $A = 1, 2, 3, \ldots, A_{max}$, the survival function counts the frequency of nodes/links that have more than $A$ degrees/lengths, $S(A) = \Sigma_{i=1}^{A_{max}} n_i$. In this work, we plot the distributions for each community separately with colours that match with the colour of communities in the shown week, see figure 1. Note that the mobility networks are directed and weighted, meaning that the links have a direction showing the origin and destination and the links may be repeated several times. In general, the shape of the distribution varies depending on inherent traits of the system and may change over time [37,38].

If communities include a few nodes/links with large degrees/lengths and many nodes/links with a few degrees/lengths, the distribution will be skewed. An extreme skewed distribution is the power-law distribution, in which the frequency of events decreases as a power of size of the events [39].[2] Power-law distributions can be described by $N(A) \sim A^{-\alpha}$, in which $\alpha$ quantifies how heavy-tailed the distribution is. On a log–log plot, a power-law distribution appears linear, and its slope is equal to $\alpha$. Power-law behaviour is also termed 'scale-free' because it follows the same relationship at all scales [39].

# 3. Results and discussion

In figure 1, we summarize COVID-19 activity in the US in the first half of 2020[3] as well as statistics of weekly aggregated mobility data. The statistics count the number of relations, number of source and target CBGs, sum of weights and the average weights of links in the networks. The first cases of COVID-19 were announced in Wuhan, China, in December 2019, and confirmation of its transmissibility put Wuhan on lockdown. Soon after, in January, additional early cases were identified around the globe, and the US government declared a public health emergency. In February of 2020, a surge in infections led to travel restrictions between countries. In figure 1, mobility data show that the number of movements was high before the declaration of lockdowns in March. While the number of CBGs that had movement from or to it, and the weight of movement was high, the average weight of CGBs overall was low, representing a wider geographical distribution of movements in more CBGs. At the beginning of March, the WHO declared COVID-19 as a pandemic. In the US, the Grand Princess cruise ship arrived on the coast of California with 21 confirmed cases.[4] On 19 March 2020, California was the first state to issue shelter-in-place orders. Residents were asked to stay home except for essential or emergency needs [40]. New York City was the second US location to experience a severe outbreak and the city implemented hard lockdowns and quarantines. Soon after, state governments like Illinois, New York, Ohio, West Virginia, Michigan, Oregon, Indiana and Minnesota issued stay-at-home orders for all residents as the number of infections and deaths began to surge. Schools were closed in Virginia, South Carolina, Ohio and Georgia. Lockdowns shuttered businesses, colleges, restaurants and public events, while people donned masks and practised social distancing. Our data show that while movements decreased, the averaged weight of CBG movements increased, representing decreased long-distance travel and increased local movements. In April, school closures extended to Indiana, Michigan and Missouri. Stay-at-home orders were then extended to additional states such as Alabama and Missouri. While governments tried to control the outbreak, people in some states such as Michigan, Ohio and Indiana protested against lockdowns. By mid-April, the restrictions put in place by states decreased movement and helped to slow the pace of new infections. In response to the perceived improvement, and under political pressure from protests and struggling businesses, state governments began to roll back restrictions in successive phases of reopening, which were declared by the federal government. These actions were both premature and harmful, and

---

[2]https://www.yalemedicine.org/news/covid-timeline,  https://www.ajmc.com/view/a-timeline-of-covid19-developments-in-2020, ttps://en.wikipedia.org/wiki/Timeline_of_the_COVID-19_pandemic_in_the_United_States_(2020)#March

[3]https://www.nytimes.com/2020/03/09/us/coronavirus-cruise-ship-oakland-grand-princess.html.

[4]https://www.nytimes.com/2020/03/19/us/California-stay-at-home-order-virus.html.

**Figure 2.** Communities of people that mostly move within the same area on (a) 23–29 February and (b) 5–11 April. Communities are shown by different colours. Colour hues represent clusters of communities that have higher mobility connections among themselves. (c) and (d) show distribution of number of in and out links to each node (degree) inside the communities of (a) and (b). (e) and (f) are the distance distribution of links inside the communities of (a) and (b). Axes are logarithmic in the distribution panels indicating that after a threshold, degrees of nodes and distances between nodes decrease by a power-law behaviour towards larger degrees and distances. (g) and (h) compare the deviation of the communities in week 23–29 February from other weeks through adjusted rank and completeness scores.

increased the distribution and number of movements, which consequently caused the disease to surge out of control throughout much of the nation.

In figure 2, we compare the fragmentation pattern of the mobility networks of the US for 23–29 February (panel *a*) and 5–11 April (panel *b*). Areas with the same colour belong to the same community and communities with the same colour hue represent clusters of communities with stronger connections. The US has five clusters in the following regions: west, north, northeast, southeast and south. Panels (*c*) and (*d*) show the degree distribution of the nodes, and panels (*e*) and (*f*) show the edges' length distribution for the communities in panels (*a*) and (*b*) (see electronic supplementary material for detail about the distribution function). Linear behaviour of the distributions in log–log axes represents the power-law nature of these distributions. National and local lockdowns in the US came into effect in March 2020 and led to dramatic reductions in movements compared with mobility patterns in February. Figure 2 shows that quarantine policies were effective at breaking up some of the connectivity between areas by reducing the size of the communities, the degree of inward and outward movements and the number of long-distance movements (which follow a power-law behaviour in range of degree of nodes/length of links). The number of detected communities in the US increased from 46 to 62 between weeks of 23–29 February and 5–11 April. Many east coast communities, especially in the New York City metropolis and the state of Florida split into smaller communities. According to panels (*c*) and (*d*), degree of movements for all CBGs in all communities from greater than 200 movements reduced to greater than 100

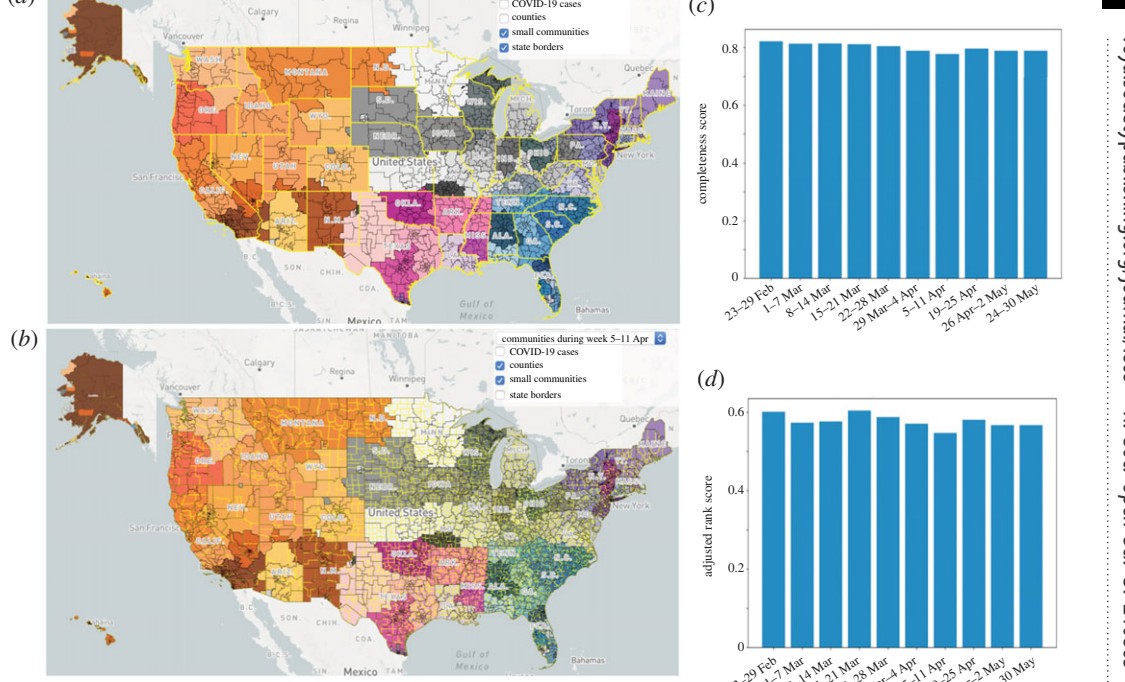

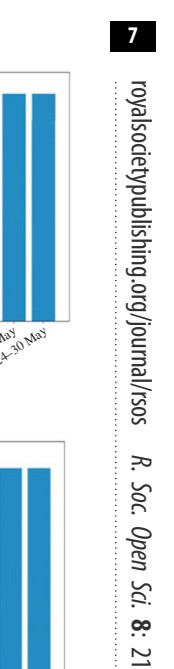

**Figure 3.** Communities (separated by different colours) and their sub-communities (separated by black lines) of the mobility pattern of the US on 5–11 April. Yellow lines show (a) state boundaries and (b) county boundaries. Deviation of communities and sub-communities from the administrative borders are clear.

movements. The highest degree of the CBGs in the communities reduced from $1500 > \text{degree} > 75\,000$ to $800 > \text{degree} > 7500$. Movement frequencies from $10^6 > F > 10^7$ decreased to $10^5 > F > 4 \times 10^6$, and the frequencies of CBG movements in all communities decreased distance due to lockdowns. For example, Florida was connected to the northeastern US cluster of communities during 23–29 February. After lockdowns were implemented in the northeast, connections became more localized and formed clusters with proximate neighbouring states. See electronic supplementary material for more details and community patterns in other weeks. In panels (g) and (h), we compare the deviations of the partitions in the first community detection level on 23–29 February from other weeks using two measures of cluster similarity: adjusted rank [41] and completeness[5] scores. These measurements evaluate the similarity of communities, with values ranging between 0 (no intersection) and 1 (perfect coherence). Our analysis shows a decrease in scores from 0.91 to 0.76 for adjusted rank and 0.81 for completeness, representing deviation from pre-existing community patterns following the implementation of lockdowns. We find the largest deviation occurred in the first week of April, when most of the country was in hard lockdown, and social distancing and mask-wearing behaviours were strongly encouraged.

Applying a community detection algorithm on the network of nodes inside each of the communities reveals the sub-structure of the communities, exposing the mobility patterns with more detail. In figure 3a,b, US mobility patterns during week 5–11 April are shown, with sub-communities separated from each other with black lines. Yellow lines represent state borders in the upper panel and county borders in the bottom panel. Although community borders align with administrative borders in some areas, in most areas community borders deviate significantly from administrative borders, see values less than 1 for adjusted rank and completeness scores, shown in panels (c) and (d), as evidence of this deviation. This indicates that policies informed by assumptions of disease containment within administrative boundaries do not account for high risk patterns of movement across and through these boundaries. Some states appear as a single community such as Michigan (MI), West Virginia (WV) and Louisiana (LA). Meanwhile, while some others merge together and appear as a large community, for example, the six-state region of New England (Maine (ME), Massachusetts (MA), New Hampshire (NH), Vermont (VT), Rhode Island (RI) and Connecticut (CT)). The North and South

[5]https://www.oneidaindiannation.com.

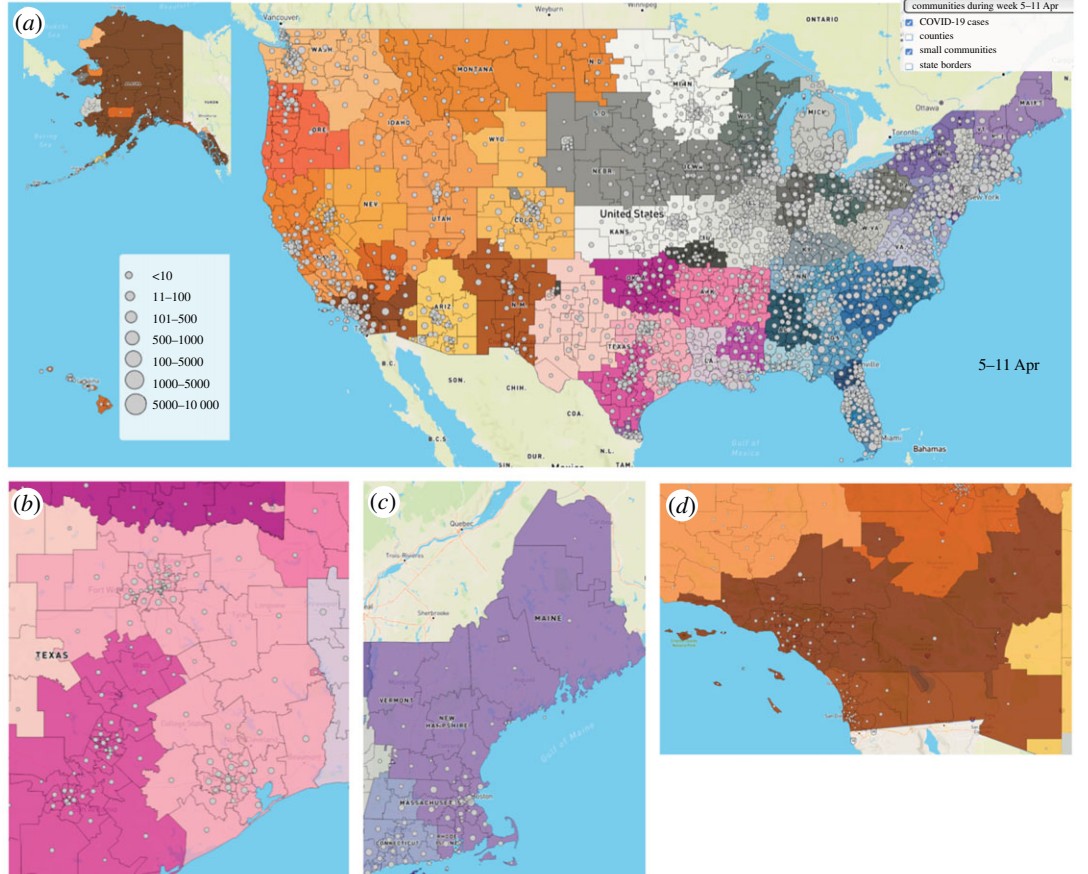

**Figure 4.** (*a*) COVID-19 cases on the map of mobility communities of the US on 5–11 April, shown by grey circles. (*b*) and (*c*) While a community appears to be in bad situation, in higher resolution (sub-communities), some of the areas are in low risk and others in high risk.

states of Carolina (NC and SC), Nebraska (NE), Iowa (IA) and South Dakota (SD) are two other examples of merged states. Some states split into small communities representing a dismantled mobility pattern within those states, such as, California (CA), Texas (TX) and Florida (FL). More intelligent contact tracing and quarantine programmes in suspected areas at the right time can dramatically slow the acceleration of the pandemic to connected areas and reduce severe impacts. Thus, it would be better to carefully define the borders of connected areas based on individuals' movements and the scale that policies are applicable and appropriate for. It is not enough to implement contact tracing that simply traces the locations where a confirmed case has been. Rather, we also need to know the locations and movements of persons who were in contact with the infected person.

Unfortunately, due to the delays in applying preventative policies across the US, many areas have seen a large number of cases. By tracking the location of recent active cases and adding them on top of the map of the communities, we can define the risk exposure for the communities. In figure 4, grey circles show the risk exposure of the communities by counting the number of active COVID-19 cases in communities and their sub-communities. While doing the analysis in lower resolutions (larger communities) can provide an aggregate view of the world situation, doing the analysis this way will mean the loss of many important details and information. For example, a community may appear to be in a bad position when it comes to COVID-19, but when we zoom into the sub-communities, we may see that there is high risk in a few particular areas, and some areas may have none demonstrating they are safer places and have a better potential to reopen earlier than higher risk areas. The higher the resolution we can provide, the better we can define the local risk levels. Communities with a higher number of confirmed cases need more extensive contact tracing and quarantine policies. Commutes from high to low risk communities can increase the spread of the COVID-19 disease across the society.

By zooming into the map (for example northeast of the US on 5–11 April, figure 5*a*), interesting facts are observable:

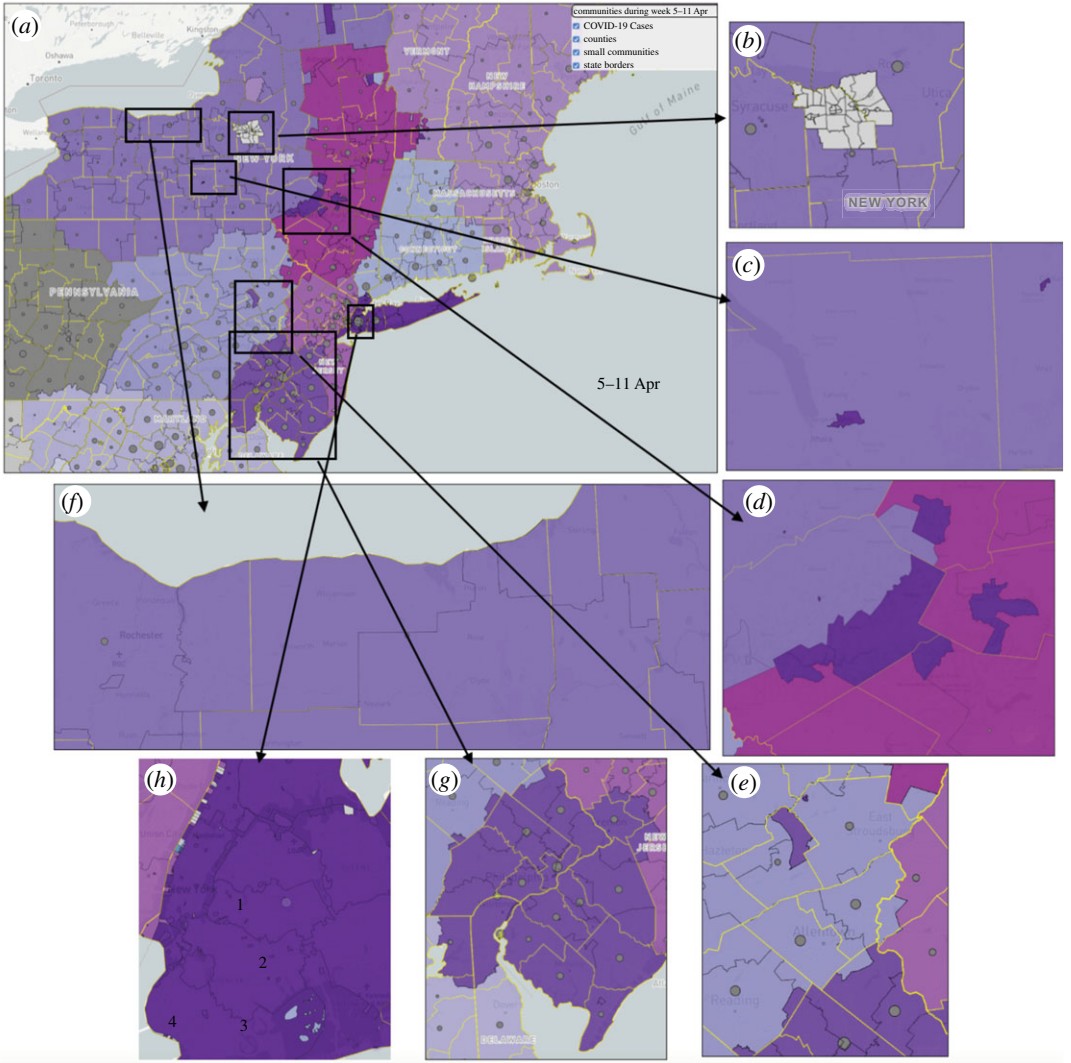

**Figure 5.** (*a*) Zoom into the northeast of the US on 5–11 April. Special examples of the map: (*b*) Areas with no mobility data, Isolated communities like (*c*) universities and (*d*) and (*e*) vacation spots, (*f*) sub-communities within other sub-communities, (*g*) communities that cross state borders and (*h*) sub-communities in city areas.

— Areas with no mobility data: There are some urban areas that do not share mobility data, like a Native American community in New York state[6], figure 5*b*.

— Isolated communities: Some parts of a community can be geographically disconnected from the rest of the community. This, for example, occurs in university and vacation areas for larger cities.

- o  Universities: New York State is the home to many universities. These universities attract people from different areas. The examples of Cornell University and SUNY Cortland, figure 5*c*, are two Universities that are located in central New York State yet are isolated sub-communities for New York City. This corresponds with a 2014 investigation,[7] which estimated that 65% of all students at Cornell from New York State came from that region of the state.

- o  Vacationers: There are vacation spots that individuals from metropolitan regions of one community go to yet are in the middle of different communities. These regions are known for their nice outdoor spaces and somewhat close proximity to the city they are connected to. This phenomenon creates isolated communities in the middle of other communities. Multiple reports have mentioned this

[6]https://ithacavoice.com/2014/11/percentage-cornell-students-come-downstate-ny/.

[7]https://www.nytimes.com/2020/03/25/nyregion/coronavirus-leaving-nyc-vacationhomes.html.

pattern occurring.[8,9] In New York City, the Catskill Mountains are one of these escapes, figure 5d, while for Philadelphia the Poconos serve the same purpose, figure 5e.

— Sub-communities within other sub-communities: All around the US, there are some areas in which people mostly interact with those immediately around them rather than their nearby urban areas. University campuses are good examples of such sub-communities. As shown in figure 5f, within the community of upstate/western New York there are specific sub-communities with connections to university campuses. On the left side of the figure, the smaller shape south of Rochester is Rochester Institute of Technology, while the right side of the figure has two smaller areas in Syracuse, both associated with Syracuse University. These universities are large and attract many students from the upstate/western New York region.

— Communities that cross state borders: Examples include the area of Philadelphia and southern New Jersey, figure 5g. The light purple region highlights the Philadelphia community. This community is multi-state and includes parts of northeastern Maryland, northern Delaware, southern New Jersey and southeastern Pennsylvania. The southern part of the Jersey Shore is a popular travel destination for people from Philadelphia, and the areas in Delaware and Maryland appear to be extensions of the great Philadelphia area.

— Sub-communities in city areas: Racial and income differences, city infrastructure and transportation can be reasons for community formation in city areas. In New York City, these communities are shown by light green in figure 5h. Brooklyn and Queens have defined sub-communities that are necessary to investigate. Sub-community 1 includes areas of Queens (Long Island City, Astoria, Sunnyside, Woodside, Jackson Heights, Elmhurst, Corona). Sub-community 2 includes parts of northern Brooklyn (Williamsburg, Greenpoint, Maspeth, Middle Village, Rego Park, Forest Hills, Bushwick, Ridgewood, Glendale). Sub-community 3 includes central Brooklyn (Clinton Hill, Bedford-Stuyvesant, Fort Greene, Prospect Heights, Crown Heights, Flatbush and Canarsie). Sub-community 4 includes areas around Prospect Park (Park Slope, Greenwood Heights, Kensington, Windsor Terrace, Prospect Lefferts Gardens). The public transportation that supports them is different for each area (within Brooklyn division as well). The Prospect Park area of Brooklyn is the most wealthy (sub-community 4, smallest subsection). The border between sub-community 4 and 3 on the map can basically be the wealth divide.[10] The racial divide between sub-community 3 and 4 can be striking on this map as well [42].

# 4. Conclusion

In conclusion, mobility patterns are one of the signs that not only reveal the effectiveness of lockdown policies, but also define areas that are in high risk with regard to the severity of COVID-19 exposure and need for more restriction actions. Mobility networks represent patches that people mostly stay within. This is important because they are also mostly in contact with individuals inside those patches. Lockdown and quarantine policies should attempt to change the mobility patterns and adapt to and strengthen borders of the patches. These policies should eliminate most of the long-distance movements and make them more localized. Patches in the city areas are mostly smaller and in the suburban or rural areas, they become larger. Quantifying movements from and to the patches and restricting commutes between low and high risk patches can be used to control the spread of coronavirus across various areas and help policymakers and governments to control the pandemic.

Our mobility and COVID case data have some limitations in the populations covered and geographical resolutions. For the privacy and security of users, SafeGraph aggregates its cell phone data into CBGs, and so these data cannot capture sub-CBG heterogeneity. Infected cases are further aggregated to the county level. This mismatch between geographical aggregations of mobility and infected cases creates some problems measuring risk of exposure in communities. To solve this, we split the infected cases of the counties that crossed several mobility patches. While these aggregations prohibit analysis at scales smaller than a certain resolution threshold, on larger scales these limitations pose no problems to our analysis. Despite these limitations, mobility data and SafeGraph data, in particular, have been used in many research projects during the recent SARS-CoV-2 pandemic [43–46].

---

[8]https://whyy.org/articles/corona-discount-as-rentals-advertise-seclusion-poconos-becomeper-capita-covid-leader/.

[9]http://storymaps.esri.com/stories/2016/wealth-divides/index.html.

[10]https://www.nytimes.com/interactive/2015/07/08/us/census-race-map.html.

In future works, we plan to study the reasons behind the formations of mobility patches by analysing demographic patterns of socio-economic and deprivation factors of society. We will also use these mobility patterns to develop meta-population stochastic models to simulate the spread of the COVID-19 virus in order to further study the effectiveness of lockdowns.

Data accessibility. Data and relevant code for this research work are stored in GitHub: https://github.com/obuchel/network_paper and have been archived within the Zenodo repository: https://zenodo.org/badge/DOI/10.5281/zenodo.5585265.svg (https://doi.org/10.5281/zenodo.5585265). Mobility data are collected by SafeGraph and shared for free for scientific researches. Other Data are available online at: https://www.endcoronavirus.org/mobility-maps.

Authors contributors. Conceptualization: L.H., Y.B.-Y, O.B. Data curation: O.B., D.C. Formal analysis: L.H., O.B. Supervision: L.H., Y.B.-Y, O.B. Writing original draft: L.H., A.N., D.C., O.B. Writing, review and editing: L.H., O.B., Y.B.-Y.

Competing interests. The authors declare no competing interests.

Funding. L.H., Y.B.-Y and O.B were supported by the US National Science Foundation (NSF) through the NSF grant no. 2032536. We thank Alex Milne for proofreading the manuscript.

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
