## [Peer Review File · Royal Society Open Science]

Review History

RSOS-210865.R0 (Original submission)

Review form: Reviewer 1

Is the manuscript scientifically sound in its present form?

No

Are the interpretations and conclusions justified by the results?

No

Is the language acceptable?

Yes

Do you have any ethical concerns with this paper?

No

Have you any concerns about statistical analyses in this paper?

Yes

Recommendation?

Major revision is needed (please make suggestions in comments)

Comments to the Author(s)

This is an interesting paper, computing communities from mobility patterns in the USA through mobile phone data. They analysed mobility network among census block and extracted communities across spatial scales by using Louvain algorithm. They focused their attention on the analysis of the overlap among administrative boundaries and the computed communities to help policymaker to put in place targeted travel restrictions. 1) I have major recommendations for quantifying the overlap among communities and administrative areas, as authors have only qualitatively described overlap on maps. 2) I suggest citing this paper: <https://journals.plos.org/plosone/article?id=10.1371/journal.pone.0081707> as they did a similar analysis in 7 countries in Europe. 3) I think also that is important a clarification on the selection of the used community detection algorithm as various algorithms exist in literature and a paragraph with the limitation of the work.

Review form: Reviewer 2

Is the manuscript scientifically sound in its present form?

No

Are the interpretations and conclusions justified by the results?

Yes

Is the language acceptable?

Yes

Do you have any ethical concerns with this paper?

No

Have you any concerns about statistical analyses in this paper?

No

Recommendation?

Major revision is needed (please make suggestions in comments)

Comments to the Author(s)

The paper addresses an important topic, the urgent issue of designing COVID-19 lockdowns. Specifically, the focus of this paper is on the effects of mobility patterns on the effectiveness of lockdown policies. To my understanding, the authors present visualizations of data on mobility patterns and COVID-19 incidence rates in the US from February and April 2020, and apply an algorithm to the mobility data to have values for how much communities move to neighbouring communities. The main argument in the paper is that controlling movements between areas depending on disease risk levels is of prime concern to curb the spread of COVID-19. I have several critical remarks/suggestions.

Abstract

- 1) Please define what the specific terms such as “social fragmentation” or “natural break point” are.
- 2) You mention that “policymakers can impose travel restrictions that are minimally disruptive to social and economic activity.” Do you have any specific findings on travel restrictions (i.e. an optimal travel restriction rate to curb the disease transmission as a function of incidence rate) to present?

Introduction

- 3) Where do you position your study in the literature? Could you please clearly state the contribution to the literature?
- 4) For an easier understanding of your work, could you concisely state what you did, what you found, how you did, why you did and why we care about your findings in one paragraph in the introduction?
- 5) Please define “fragmentation pattern”.

Materials and Methods

- 6) Could you please present summary statistics of the data sets you used? In the text, we don't have any information on crucial information such as sample size.
- 7) You explain the methods you employ piece by piece. Could you please first give an outline of the methods with links between them, for a better understanding of the paper?
- 8) Could you please explain the Louvain method?
- 9) For a reader to follow your paper easier, try to use the main verb as early as possible in a sentence. For example, on page 2, line 46, you may want to rewrite “By creating a network that consists of communities as nodes and aggregate mobilities between them as the links, we can define inter-community distances.” as “We can define inter-community distances, by creating a network that consists of communities as nodes and aggregate mobilities between them as the links.”

Results and Discussion

- 10) If you want to include the first paragraph of “Results and Discussion”, it should rather be placed in the “Introduction” in my humble opinion. It is not about your results. It provides some general background information.
- 11) Explanation of Figure 1: Is this your argument: “We have two pieces of data from before and after the quarantine policies, and sizes of communities, number of large distance movements etc. are lower after quarantine policies”. Please give information on what the quarantine policies are, and your analyses on how much these policies reduced the parameters of your concern in the areas you consider. You only present visualizations of data and some graphs from these the two periods you take into account and say that there is a difference in these parameters of concern following the implementation of policies.
- 12) Explanation of Figure 2: You have your argumentation. You should elaborate on your argumentation by connecting it to your results/analyses.

Conclusion

- 13) Some limitations of this study, suggested improvements and future direction of this work could be highlighted in the conclusion section.

Figures

- 14) Replace “fequencies” with “frequencies” in (E) and (F) of Figure 1.
- 15) In (C) and (D) of Figure 1, does “degree” mean “number of links to the nodes”?
- 16) I would update the text as “Fig. 1 represents the communities of people that mostly move within the same area on (A) February 23-29 and (B) April 5-11. Communities are shown by different colors. Darker color hues represent clusters of communities that have higher mobility connections among themselves. (C) and (D) show distributions of in and out links to the nodes inside the communities (degree distribution) of panels (A) and (B). (E) and (F) are the distance distributions of links inside the communities of panels (A) and (B). Axes are logarithmic in the distribution panels indicating that after a threshold, degrees of nodes and distances between nodes decrease by a power-law behavior towards larger degrees and distances.”

Decision letter (RSOS-210865.R0)

Dear Dr Hedayatifar

The Editors assigned to your paper RSOS-210865 "Strategizing COVID-19 Lockdowns Using Mobility Patterns" have now received comments from reviewers and would like you to revise the paper in accordance with the reviewer comments and any comments from the Editors. Please note this decision does not guarantee eventual acceptance.

Please submit your revised manuscript and required files (see below) no later than 21 days from today's (ie 24-Aug-2021) date. Note: the ScholarOne system will 'lock' if submission of the revision is attempted 21 or more days after the deadline. If you do not think you will be able to meet this deadline please contact the editorial office immediately.

on behalf of Dr Mirco Musolesi (Associate Editor) and Marta Kwiatkowska (Subject Editor)
openscience@royalsociety.org

Associate Editor Comments to Author (Dr Mirco Musolesi):

Associate Editor: 1

Comments to the Author:

The reviewers found the paper interesting, but identified some key concerns that should be addressed in a revised version of the manuscript. I would suggest the authors to carefully consider the points raised by them and prepare a revision accordingly.

Associate Editor: 2

Comments to the Author:

The paper appears to be sufficiently rigorous to be sent to peer review.

Reviewer comments to Author:

Reviewer: 1

Comments to the Author(s)

This is an interesting paper, computing communities from mobility patterns in the USA through mobile phone data. They analysed mobility network among census block and extracted communities across spatial scales by using Louvain algorithm. They focused their attention on the analysis of the overlap among administrative boundaries and the computed communities to help policymaker to put in place targeted travel restrictions. 1) I have major recommendations for quantifying the overlap among communities and administrative areas, as authors have only qualitatively described overlap on maps. 2) I suggest citing this paper:

<https://journals.plos.org/plosone/article?id=10.1371/journal.pone.0081707> as they did a similar analysis in 7 countries in Europe. 3) I think also that is important a clarification on the selection of the used community detection algorithm as various algorithms exist in literature and a paragraph with the limitation of the work.

Reviewer: 2

Comments to the Author(s)

The paper addresses an important topic, the urgent issue of designing COVID-19 lockdowns. Specifically, the focus of this paper is on the effects of mobility patterns on the effectiveness of lockdown policies. To my understanding, the authors present visualizations of data on mobility patterns and COVID-19 incidence rates in the US from February and April 2020, and apply an algorithm to the mobility data to have values for how much communities move to neighbouring communities. The main argument in the paper is that controlling movements between areas depending on disease risk levels is of prime concern to curb the spread of COVID-19. I have several critical remarks/suggestions.

Abstract

- 1) Please define what the specific terms such as “social fragmentation” or “natural break point” are.
- 2) You mention that “policymakers can impose travel restrictions that are minimally disruptive to social and economic activity.” Do you have any specific findings on travel restrictions (i.e. an optimal travel restriction rate to curb the disease transmission as a function of incidence rate) to present?

Introduction

- 3) Where do you position your study in the literature? Could you please clearly state the contribution to the literature?
- 4) For an easier understanding of your work, could you concisely state what you did, what you found, how you did, why you did and why we care about your findings in one paragraph in the introduction?
- 5) Please define “fragmentation pattern”.

Materials and Methods

- 6) Could you please present summary statistics of the data sets you used? In the text, we don't have any information on crucial information such as sample size.
- 7) You explain the methods you employ piece by piece. Could you please first give an outline of the methods with links between them, for a better understanding of the paper?
- 8) Could you please explain the Louvain method?

9) For a reader to follow your paper easier, try to use the main verb as early as possible in a sentence. For example, on page 2, line 46, you may want to rewrite “By creating a network that consists of communities as nodes and aggregate mobilities between them as the links, we can define inter-community distances.” as “We can define inter-community distances, by creating a network that consists of communities as nodes and aggregate mobilities between them as the links.”

Results and Discussion

10) If you want to include the first paragraph of “Results and Discussion”, it should rather be placed in the “Introduction” in my humble opinion. It is not about your results. It provides some general background information.

11) Explanation of Figure 1: Is this your argument: “We have two pieces of data from before and after the quarantine policies, and sizes of communities, number of large distance movements etc. are lower after quarantine policies”. Please give information on what the quarantine policies are, and your analyses on how much these policies reduced the parameters of your concern in the areas you consider. You only present visualizations of data and some graphs from these the two periods you take into account and say that there is a difference in these parameters of concern following the implementation of policies.

12) Explanation of Figure 2: You have your argumentation. You should elaborate on your argumentation by connecting it to your results/analyses.

Conclusion

13) Some limitations of this study, suggested improvements and future direction of this work could be highlighted in the conclusion section.

Figures

14) Replace “fequencies” with “frequencies” in (E) and (F) of Figure 1.

15) In (C) and (D) of Figure 1, does “degree” mean “number of links to the nodes”?

16) I would update the text as “Fig. 1 represents the communities of people that mostly move within the same area on (A) February 23-29 and (B) April 5-11. Communities are shown by different colors. Darker color hues represent clusters of communities that have higher mobility connections among themselves. (C) and (D) show distributions of in and out links to the nodes inside the communities (degree distribution) of panels (A) and (B). (E) and (F) are the distance distributions of links inside the communities of panels (A) and (B). Axes are logarithmic in the distribution panels indicating that after a threshold, degrees of nodes and distances between nodes decrease by a power-law behavior towards larger degrees and distances.”

===PREPARING YOUR MANUSCRIPT===

Please ensure that you include an acknowledgements' section before your reference list/bibliography. This should acknowledge anyone who assisted with your work, but does not

qualify as an author per the guidelines at <https://royalsociety.org/journals/ethics-policies/openness/>.

===PREPARING YOUR REVISION IN SCHOLARONE===

- Ensure that your data access statement meets the requirements at <https://royalsociety.org/journals/authors/author-guidelines/#data>. You should ensure that you cite the dataset in your reference list. If you have deposited data etc in the Dryad repository, please include both the 'For publication' link and 'For review' link at this stage.
- If you are requesting an article processing charge waiver, you must select the relevant waiver option (if requesting a discretionary waiver, the form should have been uploaded at Step 3 'File upload' above).
- If you have uploaded ESM files, please ensure you follow the guidance at <https://royalsociety.org/journals/authors/author-guidelines/#supplementary-material> to include a suitable title and informative caption. An example of appropriate titling and captioning may be found at https://figshare.com/articles/Table_S2_from_Is_there_a_trade-off_between_peak_performance_and_performance_breadth_across_temperatures_for_aerobic_scope_in_teleost_fishes_/3843624.

Author's Response to Decision Letter for (RSOS-210865.R0)

See Appendix A.

Decision letter (RSOS-210865.R1)

Dear Dr Hedayatifar,

It is a pleasure to accept your manuscript entitled "Strategizing COVID-19 Lockdowns Using Mobility Patterns" in its current form for publication in Royal Society Open Science. The comments of the reviewer(s) who reviewed your manuscript are included at the foot of this letter.

COVID-19 rapid publication process:

We are taking steps to expedite the publication of research relevant to the pandemic. If you wish, you can opt to have your paper published as soon as it is ready, rather than waiting for it to be published the scheduled Wednesday.

This means your paper will not be included in the weekly media round-up which the Society sends to journalists ahead of publication. However, it will still appear in the COVID-19 Publishing Collection which journalists will be directed to each week (<https://royalsocietypublishing.org/topic/special-collections/novel-coronavirus-outbreak>).

If you wish to have your paper considered for immediate publication, or to discuss further, please notify openscience_proofs@royalsociety.org and press@royalsociety.org when you respond to this email.

on behalf of Dr Mirco Musolesi (Associate Editor) and Marta Kwiatkowska (Subject Editor)
openscience@royalsociety.org

Associate Editor Comments to Author (Dr Mirco Musolesi):
Comments to the Author:

The authors addressed all the concerns raised by the reviewers - I would recommend this paper for publication.

Appendix A

Journal of Royal Society of Open Science

October 10st, 2021

Dear Editors,

We herewith submit the revised manuscript “Strategizing COVID-19 Lockdowns Using Mobility Patterns” for consideration for publication in Royal Society Open Science Journal. In the revised manuscript, we addressed all referee suggestions.

We added a paragraph to the end of the introduction providing the overall paper structure and merged one paragraph from the results section to the introduction section. We carefully went through the method section and applied the suggested corrections from two reviewers. One paragraph and a figure are added to the result section to carefully explain the timeline of COVID-19 actions and cases in the US and statistics of the mobility data used in the paper. An analysis is added to quantify the overlap among mobility communities and administrative areas. The conclusion is also updated to present the limitations of the data and plans for future work. We also compared our methods with previous research as requested.

We hope that the manuscript is now suitable for publication in its current form.

Please find attached the point-by-point response to all concerns raised by the referees. Thank you very much.

Sincerely,

Leila Hedayatifar

Yaneer Bar-Yam

New England Complex Systems Institute (NECSI)

277 Broadway Cambridge, MA 02139 #

Reviewer comments to Author:

Reviewer: 1

Comments to the Author(s)

This is an interesting paper, computing communities from mobility patterns in the USA through mobile phone data. They analyzed mobility network among census block and extracted communities across spatial scales by using Louvain algorithm. They focused their attention on the analysis of the overlap among administrative boundaries and the computed communities to help policymaker to put in place targeted travel restrictions.

We thank the referee for a careful reading of the manuscript.

1) I have major recommendations for quantifying the overlap among communities and administrative areas, as authors have only qualitatively described overlap on maps.

We measured the similarity in community patterns between weeks with the first week (February 23-29, before lockdowns) in Figure 2 and with administrative borders in Figure 3. Explanations about the added analysis are added to the paragraphs that explain Figures 2 and 3 in the Result section.

2) I suggest citing this paper: <https://journals.plos.org/plosone/article?id=10.1371/journal.pone.0081707> as they did a similar analysis in 7 countries in Europe.

The paper is added in the References and cited in the introduction.

3) I think also that is important a clarification on the selection of the used community detection algorithm as various algorithms exist in literature and a paragraph with the limitation of the work.

Thanks for your valuable comment. The Methods subsection in the part of Community Detection Algorithm is updated to explain more details about community detection approaches and clarification on the selection of the used method. We also updated the Conclusion section to address the limitations of the work.

Reviewer: 2

Comments to the Author(s)

The paper addresses an important topic, the urgent issue of designing COVID-19 lockdowns. Specifically, the focus of this paper is on the effects of mobility patterns on the effectiveness of lockdown policies. To my understanding, the authors present visualizations of data on mobility patterns and COVID-19 incidence rates in the US from February and April 2020, and apply an algorithm to the mobility data to have values for how much communities move to neighbouring communities. The main argument in the paper is that controlling movements between areas depending on disease risk levels is of prime concern to curb the spread of COVID-19.

We thank the referee for a careful reading of the manuscript.

I have several critical remarks/suggestions.

Abstract

1) Please define what the specific terms such as “social fragmentation” or “natural break point” are.

Thanks for your comment. In this work, “Social fragmentation” refers to **dismantled community mobility structures** within US society, and “natural break points” term refers to the patches made from natural mobility preferences and patterns. The following sentences in the abstract are updated to clarify these two terms:

“Here, the dynamics of **dismantled community mobility structures** within US society during the COVID-19 outbreak are analyzed by applying the Louvain method with modularity optimization to weekly datasets of mobile device locations. Our networks are built based on individuals’ movements in the US from February to May 2020. In a multi-scale community detection process using the locations of confirmed cases, natural break points **from mobility preferences and patterns** as well as high risk areas for contagion are identified at three scales.”

2) You mention that “policymakers can impose travel restrictions that are minimally disruptive to social and economic activity.” Do you have any specific findings on travel restrictions (i.e. an optimal travel restriction rate to curb the disease transmission as a function of incidence rate) to present?

We are developing a model to quantify the socio-economic effects of the lockdowns using optimized lockdown strategies considering mobility patterns. We kept the results for another paper. To not make any ambiguity in the results of the paper, we deleted the mentioned sentence from the abstract.

Introduction

3) Where do you position your study in the literature? Could you please clearly state the contribution to the literature?

We reviewed more literature and clarified the novelty of our approach in the last two paragraphs of the Introduction section.

4) For an easier understanding of your work, could you concisely state what you did, what you found, how you did, why you did and why we care about your findings in one paragraph in the introduction?

Thanks for your valuable comment. One paragraph is added to the end of the Introduction section to answer this question.

5) Please define "fragmentation pattern".

To clarify the definition of fragmentation pattern, the following sentences are added to the third paragraph of the method section:

The term network fragmentation is often used to describe network dismantling in the literature (20,21). As used in other works that employ community detection algorithms such as the Girvan-Newman method (23), here, the "Social fragmentation" term is used to represent the modular structure of a social network in the absence of links.

Materials and Methods

6) Could you please present summary statistics of the data sets you used? In the text, we don't have any information on crucial information such as sample size.

A figure (Fig.1) is added to summarize the statistics of the data and a paragraph is added at the beginning of the Results and Discussion section to explain the information in detail.

7) You explain the methods you employ piece by piece. Could you please first give an outline of the methods with links between them, for a better understanding of the paper?

The Methods section is updated and one paragraph is added to the beginning of the section.

8) Could you please explain the Louvain method?

Louvain method is explained by more details in the Methods subsection.

9) For a reader to follow your paper easier, try to use the main verb as early as possible in a sentence. For example, on page 2, line 46, you may want to rewrite "By creating a network that consists of communities as nodes and aggregate mobilities between them as the links, we can define inter-community distances." as "We can define inter-community distances, by creating a network that consists of communities as nodes and aggregate mobilities between them as the links."

Thanks for your comment. The mentioned sentence is updated. We asked a native English to read and edit the paper for other corrections.

Results and Discussion

10) If you want to include the first paragraph of "Results and Discussion", it should rather be placed in the "Introduction" in my humble opinion. It is not about your results. It provides some general background information.

The first paragraph of the Results and Discussion section is merged to Introduction as referee suggested.

11) Explanation of Figure 1: Is this your argument: "We have two pieces of data from before and after the quarantine policies, and sizes of communities, number of large distance movements etc. are lower after quarantine policies". Please give information on what the quarantine policies are, and your analyses on how much these policies reduced the parameters of your concern in the areas you consider. You only present visualizations of data and some graphs from these the two periods you take into account and say that there is a difference in these parameters of concern following the implementation of policies.

Thanks for your comment. Note that Figure 1 in the current version is Figure 2. We updated the paragraph explained this figure by adding more details about the analysis we did. We also added panels G and H to quantify the overlaps of the communities in week February 23-29 to the other analyzed weeks.

12) Explanation of Figure 2: You have your argumentation. You should elaborate on your argumentation by connecting it to your results/analyses.

Figure 2 is changed to be Figure 3. A few sentences are added to the paragraph that explains Figure 3 to connect our argumentation by presented results in the figure.

Conclusion

13) Some limitations of this study, suggested improvements and future direction of this work could be highlighted in the conclusion section.

Following sentences are added to conclusion section to address limitations of the work and future directions of the study.

"We need to mention that our mobility and COVID case data have limitations in the covered populations and geographical resolutions. For the safety of users, SafeGraph aggregated its cell phone data to CBGs, so these data cannot capture sub-CBG heterogeneity. The COVID-19 infected cases are also aggregated to the county level. This mismatch between geographical aggregations of the mobility and infected cases create some problems in the borders of the communities. To solve this problem, we had to split the infected cases of the counties that crossed several mobility patches. While all these aggregations do not allow us to go beyond a specific small scale, in large scales these limitations do not make a serious problem in our analysis. Also, despite these limitations, mobility data and SafeGraph data in particular have been used in many researches during the recent SARS-CoV-2 pandemic (33-37). In the future works, we plan to study the reasons behind the formations of mobility patches by analyzing demographic patterns of socio-economic and deprivation factors of the society. We will also use these mobility patterns to develop meta-population stochastic models to simulate spread of the COVID-19 virus and study effectiveness of lockdowns."

Figures

14) Replace "fequencies" with "frequencies" in (E) and (F) of Figure 1.

It is corrected.

15) In (C) and (D) of Figure 1, does "degree" mean "number of links to the nodes"?

Figure 1 in the revised version is changed to Figure 2. Degree refers to the number of in and out links to each node. It is clarified in the text and caption of Figure 2.

16) I would update the text as "Fig. 1 represents the communities of people that mostly move within the same area on (A) February 23-29 and (B) April 5-11. Communities are shown by different colors. Darker color hues represent clusters of communities that have higher mobility connections among themselves. (C) and (D) show distributions of in and out links to the nodes inside the communities (degree distribution) of panels (A) and (B). (E) and (F) are the distance distributions of links inside the communities of panels (A) and (B). Axes are logarithmic in the

distribution panels indicating that after a threshold, degrees of nodes and distances between nodes decrease by a power-law behavior towards larger degrees and distances.”

Caption of the figure is updated as suggested.